# Green Synthesis of the Effectively Environmentally Safe Metakaolin-Based Geopolymer for the Removal of Hazardous Industrial Wastes Using Two Different Methods

**DOI:** 10.3390/polym15132865

**Published:** 2023-06-28

**Authors:** Doaa A. Ahmed, Morsy A. El-Apasery, Amal A. Aly, Shereen M. Ragai

**Affiliations:** 1Chemistry Department, Faculty of Women for Arts, Science and Education, Ain Shams University, Cairo 11757, Egypt; sherenmohamed@hotmail.com; 2Dyeing, Printing and Textile Auxiliaries Department, Textile Research and Technology Institute (TRT), National Research Centre, 33 El Buhouth St., Cairo 12622, Egypt; elapaserym@yahoo.com; 3Pretreatment and Finishing of Cellulosic Based Textiles Department, Textile Research and Technology Institute (TRT), National Research Centre, 33 El Buhouth St., Cairo 12622, Egypt; amalahmedali@yahoo.com

**Keywords:** reactive black 5 dye, Cd^2+^, Pb^2+^, solidification, adsorption, geopolymer cement, metakaolin, cement kiln dust

## Abstract

Untreated wastewater pollution causes environmental degradation, health issues, and ecosystem disruption. Geopolymers offer sustainable, eco-friendly alternatives to traditional cement-based materials for wastewater solidification and removal. In this study, we investigate how wastewater containing organic and inorganic pollutants can be removed using geopolymer mixes based on metakaolin incorporation with cement kiln dust as an eco-friendly material. The present investigation compares the efficacy of two different techniques (solidification and adsorption) for reducing dye contaminants and heavy metals from wastewater using a geopolymer based on metakaolin incorporation with cement kiln dust. This study investigated the adsorption capacity of a geopolymer based on metakaolin incorporating two different ratios (20% and 40% by weight) of cement kiln dust (MC1 and MC2) for the reactive black 5 dyeing bath effluent (RBD) only and in a combination of 1200 mg/L of Pb^2+^ and Cd^2+^, each separately, in aqueous solutions under different adsorption parameters. The results of the adsorption technique for the two prepared geopolymer mixes, MC1 and MC2, show that MC1 has a higher adsorption activity than MC2 toward the reactive black 5 dyeing bath effluent both alone and in combination with Pb^2+^ and Cd^2+^ ions separately. The study also looked at using MC1 mix to stabilize and solidify both the dyeing bath effluent alone and its combination with 1200 mg/L of each heavy metal individually inside the geopolymer matrix for different time intervals up to 60 days of water curing at room temperature. The geopolymer matrix formed during the process was analyzed using FTIR, SEM, and XRD techniques to examine the phases of hydration products formed. The results showed that MC1 effectively adsorbs, stabilizes, and solidifies the dying bath effluent for up to 60 days, even with high heavy metal concentrations. On the other hand, geopolymer mixes showed an increase in mechanical properties when hydration time was increased to 60 days. According to our findings, the type of geopolymer developed from metakaolin and 20 wt.% cement kiln dust has the potential to be employed in the treatment of wastewater because it has good adsorption and solidification activity for the reactive black 5 dye effluent alone and for a mixture of dye pollutants with both Pb^2+^ and Cd^2+^ ions separately. Our results have significant implications for wastewater treatment and environmental remediation efforts, as they offer a sustainable solution for managing hazardous waste materials.

## 1. Introduction

The availability and quality of the world’s water resources have recently been declining because of population growth and fast industrialization [1]. Organic dye effluents and inorganic heavy metal discharges can be seen as two of the most dangerous pollutants out of a wide range of pollutants, primarily because of their toxic properties [2,3,4]. This highlights the urgent need for effective water treatment technologies that can remove these pollutants and ensure the availability of safe drinking water for the growing population.

Heavy metals in industrial wastewater are especially upsetting because they must be properly treated before being discarded and because of their harmful nature [5]. The most dangerous heavy metals, according to the United States Environmental Protection Agency (USEPA), are arsenic (As), copper (Cu), mercury (Hg), nickel (Ni), cadmium (Cd), lead (Pb), and chromium (Cr) [6]. Heavy metal ions can be classified into essential and non-essential categories. Essential ions such as cobalt (Co), chrome (Cr), nickel (Ni), zinc (Zn), iron (Fe), and manganese (Mn) are also referred to as trace elements and are needed by the body as tiny nutrients for stabilizing molecules through electrostatic attraction, participating in redox mechanisms, acting as catalysts in enzymatic processes, and keeping the balance of osmosis [7]. Contrarily, non-essential ions such as Cd, Hg, and Pb do not cause biological reactions and, even at extremely low concentrations, are detrimental to the organism [8,9]. As indicated above, As, Cu, Hg, Ni, Cd, Pb, and Cr belong to the most dangerous heavy metals, according to the USEPA [6,10]. Because of their inorganic structure, these ions do not naturally decompose and remain stable in surroundings, increasing the possibility that they may accumulate in living things through the food cycle as dangerous and cancer-causing compounds [11]. These heavy metals can cause serious health problems if they are released into the environment without proper treatment. Therefore, these heavy metal ions must be eliminated from the wastewater before it is released into the environment.

Dyes are also complex organic nonbiodegradable compounds with excellent chemical stability and water solubility. Dyes are widely used in various industries such as textiles, leather, and food [12]. There are three types of dyes: cationic, anionic, and non-ionic dyes. Among them, cationic dyes are more hazardous than anionic and nonionic dyes because they are more difficult to decompose and can go across the food chain [13]. Exposure to these harmful substances can cause various health problems such as cancer, skin irritation, respiratory issues, and neurological disorders [2,14].

The effects of dye wastewater on the aquatic ecology have made it a significant problem for clean water sources in recent years. Dye effluent discharge into the environment without treatment has substantial, irreversible negative consequences on aquatic life as well as human health. Specifically, dyes and the byproducts of their decomposition raise the need for biochemical and chemical oxygen and reduce sunlight transmission in aqueous conditions. Additionally, they include a variety of dangerous inorganic and organic chemicals, including aromatic compounds and heavy metals (lead, chromium, mercury, cadmium, and arsenic) [15,16,17]. Therefore, it is important to regulate the use and disposal of dyes to minimize their impact on human health and the environment. As a result, the disposal of dye-containing wastewater poses a significant environmental challenge owing to its toxic nature.

Many different water remediation technologies, such as chemical precipitation [18], electrocoagulation, ion exchange, membrane filtration, biological methods, sophisticated oxidation processes, and adsorption [19,20,21] have been tried up until now for the removal of heavy metals and dyes. Adsorption has been suggested because of the benefits of simple operation, affordability, and a wide variety of adsorbent materials being readily available [22,23]. Activated carbon (AC), clay minerals, chitosan, lignin, and geopolymer are some examples of adsorbents.

Geopolymers, commonly referred to as inorganic polymers, are three-dimensional amorphous or semi-crystalline alumino-silicate formations. They may be produced typically by alkaline activation and sometimes by acid activation of an alumino-silicate predecessor at room temperature or slightly above [2]. Additionally, geopolymers have received a lot of interest recently, especially in wastewater treatment for the removal of different kinds of contaminants, including heavy metals and dyes [3,24,25]. On the other hand, geological materials such as kaolin, metakaolin, and dolomite, as well as waste from industries such as slag, fly ash (FA), and sludge, can be used to create geopolymers as predecessors to alumino-silicates. Metakaolin (MK) is a frequently used alumino-silicate material for the production of geopolymer-based adsorbents because it has special adsorption qualities such as different structural selectivity, optimal sorption capacity, and cation exchangeable characteristics for different metal cations, which can be employed to improve the design of the processes of the treatment of waste water [26]. However, MK, a natural material, is a promising alumino-silicate material since it includes a variety of constituents that increase the adsorption performance of the geopolymer adsorbent, depending on the processing and initial material characteristics. [27,28]. Cement kiln bypass dust (CKD) is a fine powder that resembles Portland cement and ranges in color from gray to tan. It is a byproduct of the cement industry that is collected from electrostatic precipitators utilizing the dry process in large quantities. Several studies [29,30,31] examined the impact of CKD addition on the mechanical characteristics of the metakaolin-based geopolymer pastes. On the other hand, cementation materials or geopolymer binders were used in a few studies to study an innovative and efficient process of solidifying and removing reactive organic dye pollutants and heavy metals [32,33,34,35,36,37,38,39,40,41]. Our most recent investigations focused on the use of metakaolin as an adsorbent when combined with industrial wastes such as fly ash and ground granulated blast furnace slag for reactive dye removal [42,43,44,45,46,47].

The preparation, adsorption, and solidification properties of geopolymers based on metakaolin/cement kiln dust have received little attention recently. Here, we look at the environmental benefits of using this geopolymer mix as an adsorbent for organic and inorganic contaminants in wastewater treatment. Figure 1 depicts a schematic of the geopolymerization process used in our study to dispose of industrial waste in an ecologically responsible manner. In the current study, we evaluated how the incorporation of different ratios of cement kiln dust influenced the adsorption capacity of the metakaolin/cement kiln dust-based geopolymers. Two geopolymer mixes containing 20% CKD and 40% CKD (MC1 and MC2) were prepared and tested for adsorption on a reactive black 5 dyeing bath effluent and a combination of dye effluent and 1200 mg/L cadmium and lead ions separately. In addition, we examined the solidification/stabilization behavior of the geopolymer mix (80% MK + 20% CKD) for the removal of the reactive black 5 dyeing bath effluent, Cd^2+^ and Pb^2+^ ions (MC1-BD, MC1-BD-Cd^2+^, and MC1-BD-Pb^2+^). The leaching % values of the reactive black 5 dyeing bath effluent and both heavy metals in leachate solutions indicated that the geopolymer matrix almost completely removed all waste. The hydration product formed was not impacted by the presence of dye or heavy metals, according to the results of XRD and FTIR analysis. SEM analysis, on the other hand, showed that lead- and cadmium-insoluble compounds coated the surface of the geopolymer particles. The comparison between the results of the two methods revealed that the geopolymer mix MC1 was more efficient in adsorbing and solidifying the dye effluent and heavy metals. This finding is significant as it suggests a potential solution for industrial waste management. The use of geopolymer mix MC1 can reduce the amount of harmful pollutants released into the environment, contributing to a cleaner and safer ecosystem.

## 2. Materials and Experimental Techniques

### 2.1. Materials

The materials used in this study to prepare metakaolin/cement kiln dust-based geopolymer (MC) were metakaolin (MK), cement kiln dust (CKD), in addition to sodium hydroxide (NaOH) and liquid sodium silicate (Na_2_SiO_3_), which served as the alkaline activator. Furthermore, the salts of heavy metals used in this research were lead nitrate anhydrous and cadmium chloride dihydrate. The metakaolin chosen for the study was donated by the Hemts Construction Chemical Company in Cairo, Egypt. The cement kiln bypass dust (CKD) was produced by the EL Nahda Cement Factory in Qena, Egypt. The chemical company EL-Goumhouria in Cairo, Egypt, supplied the sodium hydroxide, while the commercial liquid sodium silicate was provided by Silica Egypt Company, Burg Al-Arab, Alexandria, Egypt, and the silica modulus SiO_2_/Na_2_O equaled 2.80. Additionally, Merck (Darmstadt, Germany) offered the 99% pure lead nitrate anhydrous and cadmium chloride dihydrate that were utilized in this investigation. For raw materials, Table 1 displays the percentage of chemical oxides.

The reactive black 5 dyeing bath effluent was utilized for the decolorization and solidification studies. The structure of this dye is shown in Figure 2.

### 2.2. Experimental

#### 2.2.1. Dyeing Baths Effluents

The dyeing wastewater was obtained after the process of dyeing the wool once, and then the dyeing wastewater was reused a second time for dyeing the wool as well. The remaining dyeing wastewater was used for the treatment process, and the concentration of the remaining dye was 35 mg/L.

#### 2.2.2. Adsorption Approach

##### (A) Preparation of MK/CKD-Based Geopolymer Mixes

For the manufacture of geopolymer pastes (MC1 and MC2), metakaolin (MK) was first thoroughly mixed with various ratios of cement kiln dust (CKD) as listed in Table 2 in a dry environment until entirely homogenous. Liquid sodium silicate and sodium hydroxide pellets were combined in a 15:15 weight percent solids ratio to form the alkaline activator solution [48]. Dry components and an alkaline activator solution were combined to form homogenous geopolymer pastes. We used a Standard Vicat device to verify the geopolymer pastes’ water consistency after thorough mixing [49]. The newly formed pastes were then placed in stainless steel, one-inch diameter cubic molds. The molds were then held at 60 °C for the first 24 h with a relative humidity of 100% to achieve the final setting and hardening. The cubes were de-molded after molding and soaked for 7 days at 100% relative humidity. The test cubes were collected, broken, and added to the stopping hydration solution of (1:1) ethanol and acetone. For 30 min, the mixture was stirred on a magnetic electrical stirrer, and then the residue was filtered, ethanol-washed, and dried for 24 h at 50 °C [50]. The dry samples were placed in a desiccator after being ground to a mean particle size of 100 μm. The compositions of the various mixes in addition to the liquid/solid ratio that gave standard consistency are given in Table 2.

##### (B) Preparation of Heavy Metals Solutions

For each 50 mL of the reactive black 5 dyeing effluent solution used in the adsorption test, we calculated x gm from each cadmium chloride monohydrate and lead nitrate salt, which gave us a total concentration of 1200 mg/L.

##### (C) Adsorption Test

The effects of the MK contents of the metakaolin/CKD-based geopolymer, adsorbent dosage, pH value, initial concentration of the reactive black 5 dye effluent (BD), presence of heavy metal (Pb^2+^ and Cd^2+^), and contact time on the adsorption properties of the metakaolin incorporated with cement kiln dust-based geopolymer were studied. In all, 50 mL of dyeing bath effluent solution with concentration (35 mg/L) and a specific weight of the adsorbent (0.01–0.1 g) were mixed together in the water bath at 140 rpm and 30 °C at different contact times and pH values [39,40,42,43,44,45,46,47]. The optimum pH, contact time, and adsorbent dosage that gave the best removal efficiency for the reactive black 5 dyeing bath effluent were measured. In addition, 50 mL from a combination of the reactive black 5 dyeing bath effluent and 1200 mg/L from each Cd^2+^ and Pb^2+^ ion separately were mixed together with the adsorbent at its optimum condition. A residual of the sample solutions could be established via filtration. We identified the absorbance (at λ max = 600 nm for reactive black 5 dye) using a Shimadzu spectrophotometer and then calculated the solution concentration from the calibration curve. Equations (1) and (2) were used to calculate the removal efficiency %, which showed how much dye and heavy metal had been adsorbed onto the geopolymer material [51].
qe = (Co − C) V/W(1)
where Co and C are the initial and equilibrium liquid-phase concentrations (mg/L), respectively; V is the volume of solution (L) and W the weight of the adsorbent (g).
Removal efficiency % = 100 (qe/Co)(2)

#### 2.2.3. Solidification/Stabilization Approach

##### (A) Preparation of MK/CKD-Based Geopolymer Samples:

To create the geopolymer pastes (MC1, MC1-BD, MC1-BD-Cd^2+^, and MC1-BD-Pb^2+^), 80% metakaolin (MK) was thoroughly mixed with 20% cement kiln dust (CKD) in a dry environment until fully homogeneous. The separate alkaline activator solutions in the absence and presence of the reactive black 5 dye effluent (MC1-BD), dye effluent + Pb^+2^ ions (MC1-BD-Pb^2+^), and dye effluent + Cd^2+^ ions (MC1-BD-Cd^2+^) were generated. First, a uniformly distributed gel was created by effectively combining liquid sodium silicate and sodium hydroxide pellets in a 15:15 weight % ratio with the solid component. We replaced the mixing water added to prepare the alkaline activator (which gave a homogenous paste with a dry component) with 100 mL from the reactive black 5 dyeing bath effluent in the case of preparation of the MC1-BD mix. Additionally, in order to make the activator for the MC1-BD-Cd^2+^ and MC1-BD-Pb^2+^ mixes, we employed 100 mL of a dyeing bath effluent combined with X gram of weight from each of the heavy metal salts individually, which gave a total concentration equal to 1200 mg/L from a total volume of activator [32,36,41]. Dry components were combined with the suitable alkaline activator solutions to create different geopolymer mixes at room temperature. Once the geopolymer pastes were thoroughly mixed, the water consistency of the mixtures was checked using Standard Vicat equipment [49]. The geopolymer mixtures were then placed in one-inch stainless steel cubic-shaped molds, and their surface was smoothed with a thin-edged trowel. The geopolymer pastes were immediately dried for 24 h at 60 °C with 100% relative humidity after molding. The components of each mix, in addition to the water/solid ratio that provided standard consistency, are listed in Table 2.

##### (B) Curing of the Geopolymer Cement Mixes:

For both mechanical and leaching testing of the different geopolymer mixes (MC1, MC1-BD, MC1-BD-Cd^2+^, and MC1-BD-Pb^2+^), selected cubes from each geopolymer mix were cured in 100 mL of distilled water for different periods up to 60 days [32,36,41]. After each hydration interval, the cubes were removed from their curing condition, and the compressive strength, total porosity %, and leaching % for the wastes (dyeing bath effluent and heavy metals) were evaluated.

##### (C) Investigation Techniques

**1.** Leaching test for soaking solutions of different geopolymer pastes

A leaching test can measure the quantity of dye and heavy metals released into the leachate solution during the curing treatment of geopolymer cubes at different hydration ages. An atomic absorption spectrophotometer (Savant AA-GBC Scientific Equipment, Australia) was employed to determine the quantities of cadmium and lead ions present in the leachate solutions. The quantity of the reactive black 5 dye effluent that remained in the leachates after different hydration period was additionally recorded using a spectrophotometer (V-670). Equation (3) was used to evaluate the leaching percentage (% L) of the reactive black 5 dyeing bath effluent, cadmium, and lead present in the leachate solution [35].
Leaching % = (C_L_/C_T_) × 100(3)
where CT is the total concentration (mg/L) of the organic dye effluent or heavy metals incorporated in the geopolymer mix cube and CL is the concentration of the dye or heavy metal ions leached out (mg/L) of the geopolymer mix cube.

To evaluate how much reactive black 5 dyeing bath effluent, cadmium, and lead ions were immobilized and solidified in the hardened geopolymer pastes, we used Equation (4) [32,33,34,35,36,37,38,39,40,41].
Immobilization = 100 − Leaching %(4)

**2.** Determination of the mechanical properties of metakaolin/cement kiln dust-based geopolymer mixes after different hydration ages.

1—Compressive strength test:

The hardened paste’s compressive strength was calculated using a set of three cubes. The data are presented in kg/cm^2^ and represent the average of the three measurements. In the present investigation, compressive strength was determined using a manually operated compression apparatus (D550-control type, Milano, Italy). Following the compressive strength test, the stopping of hydration was performed on the crushed cubic specimens. After being crushed, the samples were collected, combined with a stopping solution that included alcohol and acetone (1:1) to avoid further hydration, and dried at 50 °C for 24 h before being saved for morphology examination (XRD, FTIR and SEM tests).

2—Total porosity test:

The process of determining the total porosity % of a geopolymer mixes involved weighing samples of dry paste suspended in air and water, designated W1 and W2, respectively, for three separate cubes. These cubes were then dried at 100 °C for around 24 h to calculate their weight in the air, designated as W3. The following equation was used to calculate the total porosity percentage (P%):P% = [(W2 − W1)/(W2 − W3)] × 100.(5)

**3.** The morphology and microstructure analysis.

1—X-ray diffraction analysis (XRD).

The X-ray diffraction analysis conducted on the Bruker D8 Discover diffractometer from Germany allowed for a thorough examination of the phase composition of the hydration products in various geopolymer mix samples. By using a Ni-filtered diffractometer and Cu-K radiation with a wavelength of 1.45 Å, along with a pixel detector set to 40 kV and 40 mA, accurate results were obtained. The data collected from this analysis revealed important information about the structure and properties of the geopolymer mix samples, including their degree of crystallinity and amorphousness.

2—The Fourier transform infrared spectroscopy (FTIR)

FTIR measurements were conducted on a Perkin Elmer 1430 infrared spectrophotometer (USA) with pellets of potassium bromide (KBr), and wave numbers ranged from 400 to 4000 cm^−1^. It was used to identify the functional groups of the hydration products produced for a chosen sample.

3—Scanning electron microscopy (SEM):

The scanning electron microphotographs were obtained with a FESEM Quanta FEG 250 (Netherlands), with its energy-dispersive X-ray analyzer (EDXA) from the USA.

## 3. Results and Discussion

### 3.1. Adsorption Properties of a Geopolymer Based on Metakaolin Incorporated with Cement Kiln Dust

#### 3.1.1. Effects of Adsorption Parameters

1—Effect of pH on removal efficiency %

Figure 3 depicts the change in the removal % of dye when reactive black 5 dye was employed in the dye stream under the pH of the adsorption bath. Metakaolin/cement kiln dust based geopolymers-treated dye at various pH ranges (2–10), it was possible to estimate the ideal pH value for geopolymer cement. The percentage of dye removal efficiency decreased with increasing pH for all wastewaters, according to the results from Figure 3. Additionally, while employing geopolymer cement based on metakaolin/cement kiln dust (MC1 and MC2), which reached their maximum values at 15.1% and 10.1%, respectively, at pH 2, the percentage removal efficiencies for reactive black 5 dye effluents were at their highest levels.

2—Effect of adsorbent weight on the removal efficiency %

Figure 4 shows how the concentration of the adsorbent affects the percentage removal efficiency. The dye effluents for reactive black 5 with pH 2 for MC1 and MC2 geopolymer mix were used to test the dye’s adsorption at various concentrations (0.01 to 0.1 g/50 mL) of geopolymer cement for 120 min. The results of Figure 4 demonstrated that as adsorbent weight is increased, removal efficiency % decreases. The greatest removal efficiency % for the MC1 geopolymer mix was 46.3%, whereas the maximum removal efficiency % for the MC2 geopolymer mix was 30.4%, both at 0.01 g/50 mL.

3—Effect of time on the removal efficiency %

It is also important to note that the treatment was carried out using predetermined process settings for varying times (60–240 min) in order to identify the ideal duration of reactive dye interaction with the slag-based geopolymer blends. The findings in Figure 5 demonstrated how lengthening the adsorption time might increase the dye’s removal effectiveness. As the time passed, the amount of the removed color increased until it reached a maximum value, at which point removal efficiency % became steady. At 120 min, the decolorization rate for the MC1 geopolymer mix was 46.3%, and for the MC2 geopolymer mix it was 30.4%.

4—Metal effect on dye adsorption % at optimum conditions

The adsorption characteristics of two prepared geopolymer mixes, MC1 and MC2, toward the dye effluent in the presence of 1200 mg/L of two different heavy metals (Cd^2+^ and Pb^2+^) at optimum conditions were also evaluated (Table 3). Table 3 also shows the adsorption rates of MC1 and MC2 mixes for lead and cadmium ions in the presence of the dye effluent. The results indicated that the presence of heavy metals had a positive impact on the adsorption of dye effluent by the MK/CKD geopolymer mixes. On the other hand, the presence of lead ions had a retardation effect on the reactive black 5 dye effluent removal by the MC1 mix (Table 3). We also noticed in Table 3 that the geopolymer mixes had a greater tendency for Pb^2+^ ion removal (47% and 53%) than for Cd^2+^ ion removal (7% and 8.3%). The presence of Cd^2+^ accelerated the adsorption of dye effluent (46.3% and 54.7%) with a combination of hardness in its adsorption on the geopolymer matrix. This may be because there were a number of active sites on the adsorbent surface, and with the increase in the adsorption of dye molecules, the number of effective adsorption sites on the adsorbent surface decreased, and the adsorption of cadmium became harder. In the case of Pb^2+^ in MC1 there was retardation in the adsorption process of the dye effluent by the active site (from 46.3%→22.9%) accompanied by increase in the lead content adsorption. This could be explained by an increase in the adsorption of lead ions on active sites leading to a reduction in the number of active sites on the surface, which retards the adsorption of dye effluent. 

#### 3.1.2. Adsorption Isotherms

In our study, we used Langmuir and Freundlich models, where each system was essential to optimizing the design of an adsorption system to remove the dye effluent on the surface of MC1 and MC2 geopolymer mixes. The Langmuir model is based on the assumption that the adsorbent surface is homogeneous, and that adsorption occurs on a monolayer. On the other hand, the Freundlich model is based on the assumption that adsorption occurs on a heterogeneous surface and that multiple layers can be formed. By using both models, we were able to determine the maximum adsorption capacity of our chosen adsorbent and how it varied with changes in concentration and temperature.

1—Langmuir isotherm

To determine the equilibrium isotherms of dye on the two adsorbents MC1 and MC2, Equation (6) from the Langmuir model was used [52].
Ce/qe = 1 /K_L_ + a_L_/K_L_ (Ce)(6)

Ce is the adsorbate’s equilibrium concentration, qe is the equilibrium amount of adsorbed dye per unit mass, and K_L_ is the Langmuir isotherm constant. The maximum adsorbent capacity for adsorbate (Q max) was determined from the ratio K_L_/a_L_. 1/K_L_ is the intercept of the plotting curve (Figure 6B and Figure 7B) plotted for (Ce/qe) of MC1 and MC2 against the remaining concentration of dye in solution [52]. a_L_/K_L_ is the slope of the plotting curve in Figure 6B and Figure 7B.

2—Freundlich isotherm

The Freundlich equation [53,54] is the most appropriate equation for clarifying sorption onto surfaces with active sites of different affinities. The following is the Freundlich Equation (7):Log qe = log K_F_ + 1/n log Ce(7)

The adsorption capacity is determined by the intercept K_F_ obtained from the plot of log qe against log Ce (Figure 6C and Figure 7C), and the beneficial and capacity nature of the adsorbent–adsorbate system is identified by the slope (1/n). According to Table 4 the n values for MC1 and MC2 were respectively 3.067485 and 3.968254, which were both greater than 1 and smaller than 10. Additionally, K_F_ values for MC1 and MC2 were 2.8314 and 1.9634, respectively, demonstrating that K_F_ > 1. The outcome then showed that the Freundlich theory was supported and improved by this adsorption interaction process on the two geopolymer mixes [55,56].

Table 4 shows K_L_, a_L_, and Q_max_ for the dye on the two geopolymer mixes. The Q_max_ values were 7.692308 and 4.524887 mg/g for MC1 and MC2, respectively, indicating the adsorption activity of MC1 was greater than that of MC2. Table 4 data show that the values of (R2) were 0.9911 and 0.9949, which lie in the range of 0 to 1 (Figure 6B and Figure 7B). This indicated that the adsorption process was favorable. In addition, the K_L_ value, which was both positive and greater than unity, showed an improved sorption affinity [53].

***According to the outcomes*** of our adsorption methodology, MC1 had a higher adsorption activity than MC2. The second section of this study, which focused on using an innovative approach for disposing of hazardous waste through solidification inside the geopolymer matrix, employed the geopolymer mix MC1.

### 3.2. Solidification/Stabilization of Heavy Metals and Dye Effluent inside MK/CKD-Based Geopolymer Mix (MC1)

#### 3.2.1. Leaching % Test Results

##### (A) For Reactive Black 5 Dyeing Bath Effluent in Different Leachate Solutions

According to UV-Vis absorbance spectra of leachate solutions after varying curing times from 1 to 60 days, Figure 8 illustrates the solidification/stabilization performance of MC1-BD, MC1-BD-Cd^2+^, and MC1-BD-Pb^2+^geopolymer mixes against absorbance bands related to the reactive black dying bath effluent (35 mg/L). The UV-Vis absorption band spectra of the reactive black 5 dye are depicted in Figure 8, where different wavelengths (312, 392, 422, and 600 nm) recorded a maximum. Absorption measurements of the leachate solutions for geopolymer mixes MC1-BD, MC1-BD-Pb^2+^, and MC1-BD-Cd^2+^ revealed a complete disappearance of the absorption band maxima related to the reactive black 5 dye after all hydration times, as we notice from Figure 8A–C. This may be clarified by the enhanced ability of the geopolymer matrix to adsorb contaminants, including heavy metals and organic dyes, inside its matrix [32,36,41]. This adsorption process occurred owing to the high surface area and porosity of the geopolymer matrix, which allowed for the effective binding of contaminants. The determined leaching percentage and immobilization of the reactive black 5 dyeing bath effluent in various leachate solutions after varying times of curing are shown in Table 5. Results from the leachate measurements (Table 5) also showed that the addition of heavy metals insignificantly affected the leaching behavior of the geopolymer mixes toward the dyeing bath effluent, indicating that geopolymerization can be an effective method for solidifying dyeing bath effluent. 

##### (B) For Pb^2+^ and Cd^2+^ Ions in Different Leachate Solutions

The performance of the solidification/stabilization processes of both cadmium and lead ions (1200 mg/L) in the presence of the reactive black 5 dye effluents inside the metakaolin/CKD-based geopolymer matrix was examined after different curing durations from 1 to 60 days. Table 5 contains the findings of the leaching % of both heavy metal ions in the various MK/CKD-based geopolymer mixes (MC1-BD-Cd^2+^ and MC1-BD-Pb^2+^). Following 60 days of hydration, it was observed that the cadmium (II) ions leaching % was equal to 0.0008621, while lead ions were equal to 0.010344, and immobilization measurements were close to 100. It may be explained by the geopolymer paste’s extraordinary ability to solidify both ions inside its matrix over the course of 60 days. The MK/CKD-based geopolymer showed exceptional adsorption capacity for lead and cadmium ions, effectively reducing their concentrations to safe levels (>1 mg/L). Additionally, the geopolymer was successful in treating reactive black 5 dyeing bath effluent, a notoriously difficult substance to remove from wastewater.

##### (C) Tests for pH of Leachate Solutions as a Function of Contact Duration

Table 6 summarizes the changes in the values of pH of curing solutions of MK/CKD-based geopolymer mixes (MC1, MC1-BD, MC1-BD-Cd^2+^, and MC1-BD-Pb^2+^) as a function of contact time from 1 h to 60 days of hydration. Hydration of geopolymer mixes over 28 days resulted in high levels of alkalinity in leachate solutions, because of the release of alkaline species into leaching solutions (Table 6). As shown in Table 6, the pH values for MC1-BD, MC1-BD-Cd^2+^, and MC1-BD-Pb^2+^ remained constant from 28 to 60 days. This was because no new hydrated compounds were formed, and it may also be that layers of compounds from heavy metals and dye effluent filled the pores on the surface of the geopolymer matrix, preventing any ions from the interior of the geopolymer matrix from escaping into the soaking solution.

According to our leaching % test results, geopolymerization is a promising solution for the safe disposal of wastewater containing hazardous substances such as heavy metals and dyes. It is a low-cost and environmentally friendly process that can be scaled up for industrial applications.

#### 3.2.2. Mechanical Properties of Various Geopolymer Mixes Based on Metakaolin Incorporated with Cement Kiln Dust (MC1) after Curing in Different Hydration Ages

##### (A) Compressive Strength Measurements

The compressive strength properties of the following MK-CKD-based geopolymer mixes (MC1, MC1-BD, MC1-BD-Cd^2+^, and MC1-BD-Pb^2+^) after different hydration periods from 1 to 60 days are depicted in Figure 9. It is clear that all the mixes’ compressive strength values rose as the curing process continued. This may be due to the interaction between metakaolin and the free lime (CH) that was released from cement kiln dust as well as the presence of alkalis in cement kiln dust, which increased the rate at which metakaolin dissolution occurred. This released silicate and aluminate species into the system, completing the geopolymerization process [31]. The possibility of forming additional hydration products, such as C-S-H, C-A-S-H, and N-(C)-A-S-H gel, which is deposited in the open pores and increases compressive strength, increases when the amount of reacted metakaolin increases [31]. The presence of the dyeing bath effluent and heavy metal ions (MC1-BD, MC1-BD-Cd^2+^and MC1-BD-Pb^2+^) enhanced the compressive strength values of the MC1 geopolymer from 1 day to 60 days of hydration, as we noticed from Figure 9. This may be attributed to the adsorption of the dye effluent at the active site and the intra-particle diffusion of the dye effluent into the interior pores of the geopolymer sorbent matrix, which filled open pores and increased the compressive strength of the geopolymer mix [57]. On the other hand, Pb^2+^ was involved in the polycondensation of the Si/Al gel phase and created PbO inside the network of the geopolymer [58], which may have caused the progress in enhancing compressive strength by adding lead (II) ions. Furthermore, the MC1-BD-Cd^2+^ geopolymer mix exhibited the highest compressive strength values, which may be attributed to the formation of Cd(OH)_2_ that was precipitated into the geopolymer matrix and filled the open pores [59].

##### (B) Total Porosity (P%) Measurements

Figure 10 demonstrates the total porosity % of the different geopolymer mix cubes MC1, MC1-BD-Cd^2+,^ MC1-BD, and MC1-BD-Pb^2+^ cured in H_2_O for 1, 3, 7, 14, 28, and 60 days. The total porosity for all geopolymer mixes was observed to decrease with curing time. This was explained by the increased pozzolanic activity of metakaolin, which combined with the lime produced from cement kiln dust to generate extra quantities of hydration products that precipitated in some accessible open pores, leading to a reduction in the total porosity % [31]. The addition of both heavy metals and dyeing bath effluent to geopolymer mixes MC1-BD-Cd^2+^ and MC1-BD-Pb^2+^ eliminated the total porosity % values at all hydration times, as shown in Figure 10. This may be due to the adsorption of the dye effluent at the active site and the intra-particle diffusion of the dye effluent into the interior pores of the geopolymer matrix [58], which led to the filling of open pores and decreased the total porosity values. Additionally, the precipitation of insoluble compounds from lead (II) and cadmium (II) inside the geopolymer matrix reduced the size of the pores and reduced the total porosity values [24,58,59].

##### (C) The Compressive Strength Mean Values:

The statistical mean values for compressive strength are displayed in Figure 11a–d. This Figure illustrates the average compressive strength values for a range of geopolymer mixes in the presence and absence of the dye effluent and heavy metals. The data points show that there was a wide variation in compressive strength across different geopolymer mixes and different hydration times, with some mixes exhibiting much higher values than others.

#### 3.2.3. The Morphology and Microstructure of Geopolymer Mixes Based on MK/CKD after Curing at Different Hydration Ages

##### (A) Fourier Transform Infrared Spectroscopy Analysis (FTIR)

The Fourier transform infrared (FTIR) analysis of the four metakaolin/cement kiln dust-based geopolymer pastes, including MC1, MC1-BD, MC1-BD-Cd^2+^, and MC1-BD-Pb^2+^, after 7 and 28 days of hydration can be observed in Figure 12. Two absorption bands in the FTIR spectrum at 1416–1439 cm^−1^ and 870–875 cm^−1^ were formed as a result of the carbonation of calcium hydroxide. By adding the reactive black 5 dyeing bath effluent and both heavy metals to the geopolymer matrix, the carbonation process was retarded, as shown by the reduction in the strength of these bands (Figure 12A,B).

These results were attributed to the positive effect of the dye effluent, Cd^2+^, and Pb^2+^ ions in the filled open pores in the geopolymer matrix and were in good agreement with the mechanical results. Additionally, we observed in Figure 12B that the carbonation process was slowed down by prolonging hydration because more hydration products were produced in the open pores as the hydration progressed. In all MK/CKD-based geopolymer systems, we also found a shift in the dominant and strong absorption band at approximately 1000 cm^−1^, which corresponded to the asymmetric stretching vibrations of Si-O-T (where T = Si or Al) to a lower wavenumber at about 960–982 cm^−1^. The band shift can be explained by the geopolymerization process and the growth of amorphous alumino-silicate gels (CSH and N-(C)-A-S-H) in geopolymer binders [60]. This was consistent with the intensity of bands at about 777 and 693 cm^−1^ that, respectively, correlated to the symmetric stretching vibrations of (Si-O-Si) and (Si-O-Si or Al-O-Si), which confirmed the full dissolution of unreacted silica and the development of the polymerization process [61]. The intensity of the asymmetric stretching vibration of the Si-O-T band at 960–965 cm^−1^ in the MC1 mix was higher than that of other mixes containing dye effluent and heavy metal ions. This may be due to the reduction in the geopolymerization rate due to the adsorption of dye molecules and the precipitation of heavy metals at the active site of the geopolymer matrix. The strongest band of the O-Si-O bending vibration, which was associated to quartz in the unreacted metakaolin, lay at 442–453 cm^−1^ [62].

##### (B) X-ray Diffraction Analysis (XRD)

Figure 13 displays the XRD patterns of the geopolymer mixes (MC1, MC1-BD, MC1-BD-Cd^2+^, and MC1-BD-Pb^2+^) following hydration for 28 days. The XRD patterns of all geopolymer samples (Figure 13) show a hug and broad hump between 22° and 38° 2θ, suggesting that the metakaolin species entirely disintegrated and dissolved as a result of alkaline activation and the creation of an amorphous geopolymer matrix [31,63]. When metakaolin dissociated, free silica was released and partially substituted with aluminum (Al) atoms or combined with calcium hydroxide in CKD, suggesting the formation of a new crystalline phase such as a (N-/C-(A-S-H)) type gel within the geopolymerization process, which was indicated by XRD patterns at d-values (12.28, 8.60, 7.09, 5.49, 3.70, 3.29, 2.61, 1.73). It was determined that the crystalline phase (CSH), with basal reflections at d = 3.03 and 2.786 Å, also formed as the other hydration result of the geopolymerization process (Figure 12). This indicated that the geopolymerization process resulted in the formation of both amorphous and crystalline phases. Figure 13 illustrates that the main phase in all diffractograms of geopolymer pastes with basal reflections (d = 3.34, 4.29, 2.28, 1.97, 1.61, 1.69, 1.54, 1.45 Å) was quartz (SiO_2_). This demonstrated that this mineral did not take part in the geopolymerization reaction and was caused by unreacted silica from the raw materials. The presence of the reactive black 5 dyeing bath effluent, cadmium ions, and lead ions (MC1-BD, MC1-BD-Cd^2+^, MC1-BD-Pb^2+^) had no effect on the nature of the hydration product produced in the geopolymer pastes, as shown in Figure 13. Figure 13 shows that the presence of heavy metal ions and dye effluent in the geopolymer mix increased the intensity of the hydration product characteristic peaks (CSH, CASH) while decreasing the intensity of the calcium carbonate characteristic peaks, which indicated a slowing of the carbonation process owing to its filled open pores. This was in good agreement with the compressive strength results.

##### (C) SEM Analysis

Figure 14, Figure 15, Figure 16 and Figure 17 illustrate the microstructures of the various prepared metakaolin/cement kiln dust-based geopolymers in the absence and presence of the dye effluent and heavy metals (MC1, MC1-BD, MC1-BD-Cd^2+^, MC1-BD-Pb^2+^) after 28 days of curing. SEM analysis revealed that the MC1 mix microstructure seemed to be less dense, with many pores and large particles dispersed throughout the geopolymer matrix, which may represent unreacted metakaolin and raw material (Figure 14). This observation was in good agreement with the geopolymer mix MC1 mechanical properties results, which indicated that MC1 had lower mechanical properties than other mixes (Figure 9 and Figure 10). On the other hand, the addition of the reactive black 5 dye effluent led to a more compact, dense, and homogeneous microstructure, as we notice from Figure 15. These data indicated that the presence of the dye effluent enhanced the amount of the activated species that condensed to form a geopolymer, filling open pores and thereby enhancing the microstructure’s densification. Figure 16 shows that the densification of the microstructure was improved also by the addition of cadmium ions along with the dye effluent to the geopolymer paste (MC1-BD-Cd^2+^). This may have been accomplished by the formation of an insoluble cadmium compound [59] that coated the geopolymer’s surface. Furthermore, the addition of lead ions combined with the dye effluent in the MC1-BD-Pb^2+^ mix improved the geopolymer’s microstructure by coating its surface and densifying the geopolymer microstructure (Figure 17), which was explained in earlier studies because lead ions and compounds are disrupted uniformly across all geopolymer chains [64,65].


**Comparison between two approaches in its environmental impact**


**1**—According to our results, the adsorption technique succeeded in removing approximately 65% of inorganic and organic wastes in solution (dye effluent and heavy metals) by 0.01 gm from both geopolymer mixes MC1 and MC2.

**MC1**-→47% Pb^2+^ + 22.9% dye effluent and 7% Cd^2+^ ions + 54.7% dye effluent

**MC2**-→53% Pb^2+^ + 37.64% dye effluent and 8.3% Cd^2+^ ions + 39.64% dye effluent.

**2**—The adsorption activity of MC1 was larger than that of MC2, as evidenced by the Q_max_ values of 7.692308 and 4.524887 mg/g for MC1 and MC2, respectively. The additional isotherm characteristics of adsorption also supported the beneficial nature of the process.

**3**—The solidification/stabilization method for geopolymer mix MC1 was efficient at removing 100% of the reactive black 5 dyeing bath effluent from the matrix after 1 day and for 60 days after water curing.

**4**—The solidification method solidified approximately 99.99914 Cd^2+^ ions and 99.98966 Pb^2+^ ions inside the MC1 matrix after 60 days of water curing (1200 mg/L from each metal/400 g from geopolymer) in the presence of dye pollutant.

**5**—The amount of heavy metals by ppm in the soaking solution in our study was less than the limits according to the Environmental Law (Decree 44/2000) for discharging treated water in industrial areas (1 mg/L for Pb^2+^ and 0.2 mg/L for Cd^2+^). Our study’s measurements revealed that after 60 days of water curing, Pb^2+^ ion concentrations were 0.36 mg/L and Cd^2+^ ion concentrations were 0.03 mg/L.

## 4. Conclusions

In the present study, innovative combinations of metakaolin and cement kiln dust that support geopolymer mixes and replace conventional cement as a sustainable alternative were suggested. Additionally, we investigated the use of a geopolymer binder in the environment application as an adsorbent for organic and inorganic pollutants to clean wastewater. The findings of the adsorption technique for the two prepared geopolymer mixes, MC1 (80% MK + 20% CKD) and MC2 (60% MK + 40% CKD), revealed that MC1 had a higher adsorption activity than MC2 toward the reactive black 5 dyeing bath effluent, both alone and in combination with each Pb^2+^ and Cd^2+^ ions separately. The Freundlich model was supported and improved by this adsorption interaction process on the two MK/CKD-based geopolymer mixes, according to the isotherm data. On the other hand, the results of the solidification test of the MC1 geopolymer cement mix in the dyeing bath effluent and heavy metals showed that the presence of dyeing bath waste alone or combined with Pb^2+^ and Cd^2+^ ions improved MC1’s mechanical characteristics. Leaching % results suggested that the prepared sustainable geopolymer mix (MC1) removed approximately 100% of the dyeing bath effluent as well as lead (II) and cadmium (II) ions. The morphology and microstructure data also proved that the presence of the dyeing bath effluent and both heavy metals improved the geopolymer matrix. These findings have important implications for removing harmful substances from wastewater, improving environmental sustainability, and using sustainable construction materials.

## Figures and Tables

**Figure 1 polymers-15-02865-f001:**
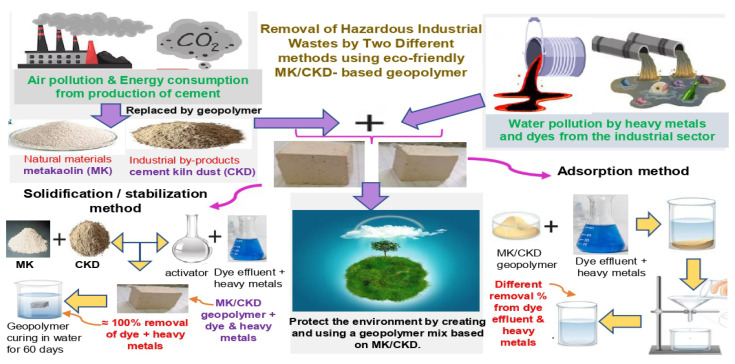
Schematic diagram of the recycling process used in our research.

**Figure 2 polymers-15-02865-f002:**
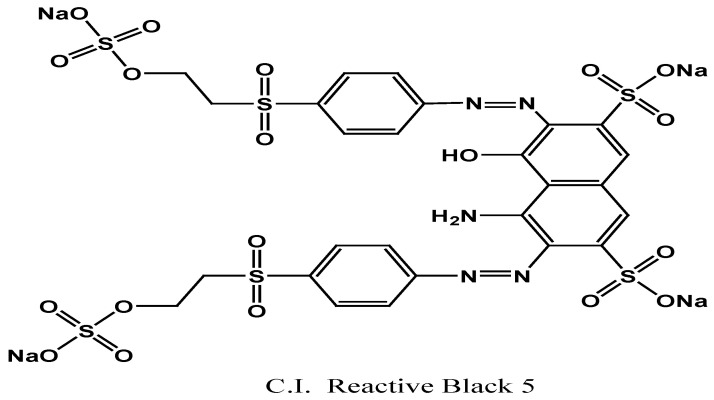
C.I. Reactive black 5.

**Figure 3 polymers-15-02865-f003:**
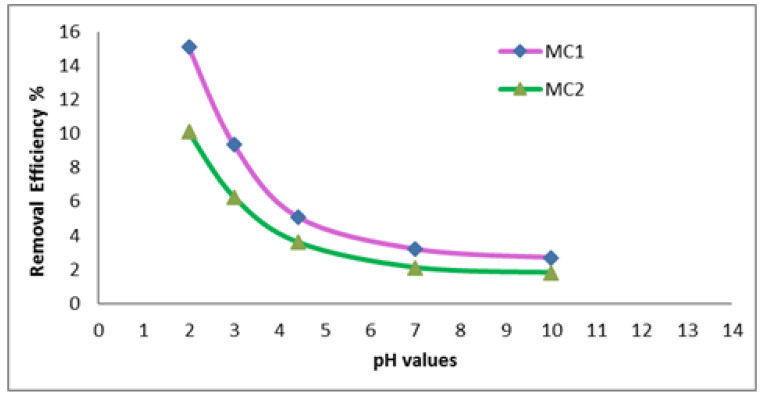
Effect of pH on dye removal efficiency % (Time 120 min, temperature 30 °C, wt. of adsorbent 0.03 g, concentration of dye 35 mg/L).

**Figure 4 polymers-15-02865-f004:**
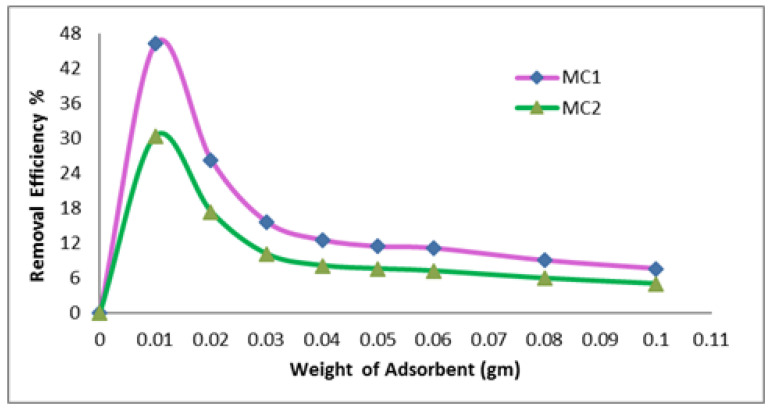
Effect of adsorbent weight on the removal efficiency % (Time 120 min, temperature 30 °C, pH 2, concentration of dye 35 mg/L).

**Figure 5 polymers-15-02865-f005:**
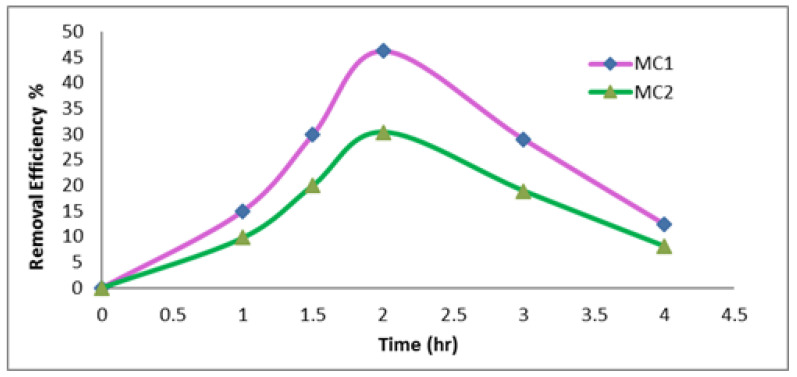
Effect of time on the removal efficiency % (wt. of adsorbent 0.01 g, temperature 30 C°, concentration of dye 35 mg/L, pH 2).

**Figure 6 polymers-15-02865-f006:**
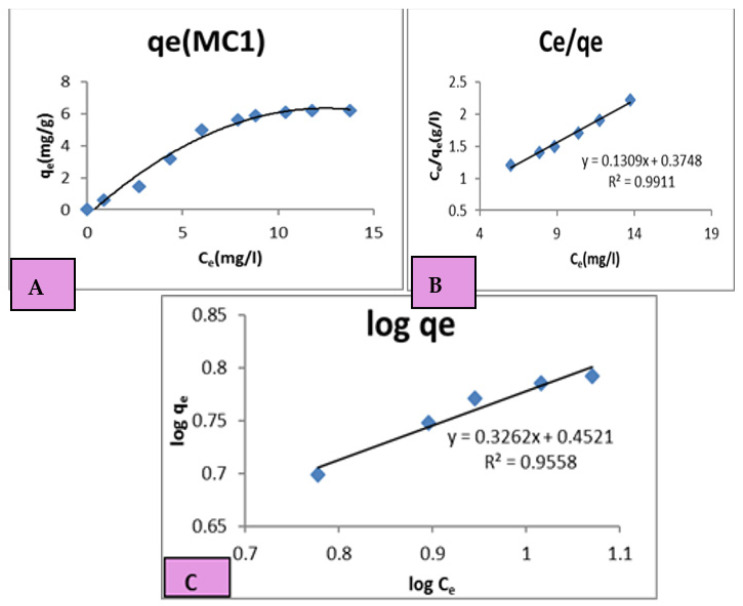
Kinetic models for the adsorption process onto the MC1 geopolymer mix: (**A**) Isotherm for MC1, (**B**) Langmuir Model for MC1, (**C**) Freundlich Model for MC1.

**Figure 7 polymers-15-02865-f007:**
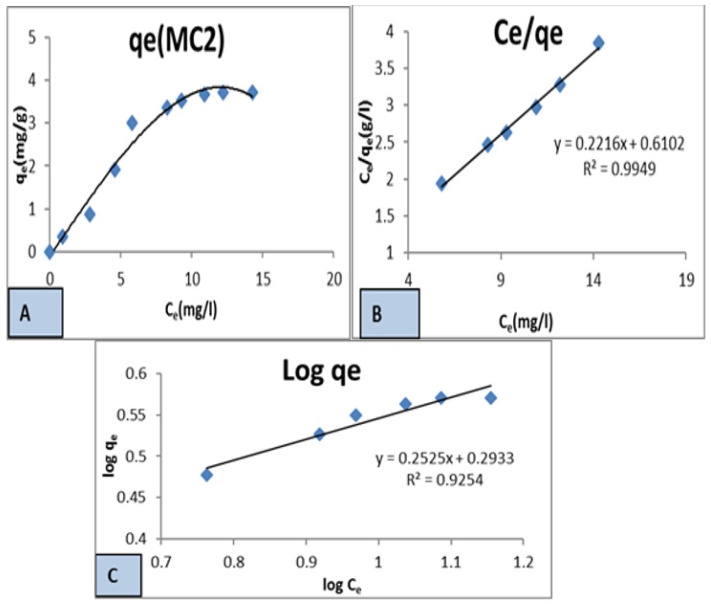
Kinetic models for the adsorption process onto the MC2 geopolymer mix: (**A**) Isotherm for MC2, (**B**) Langmuir Model for MC2, (**C**) Freundlich Model for MC2.

**Figure 8 polymers-15-02865-f008:**
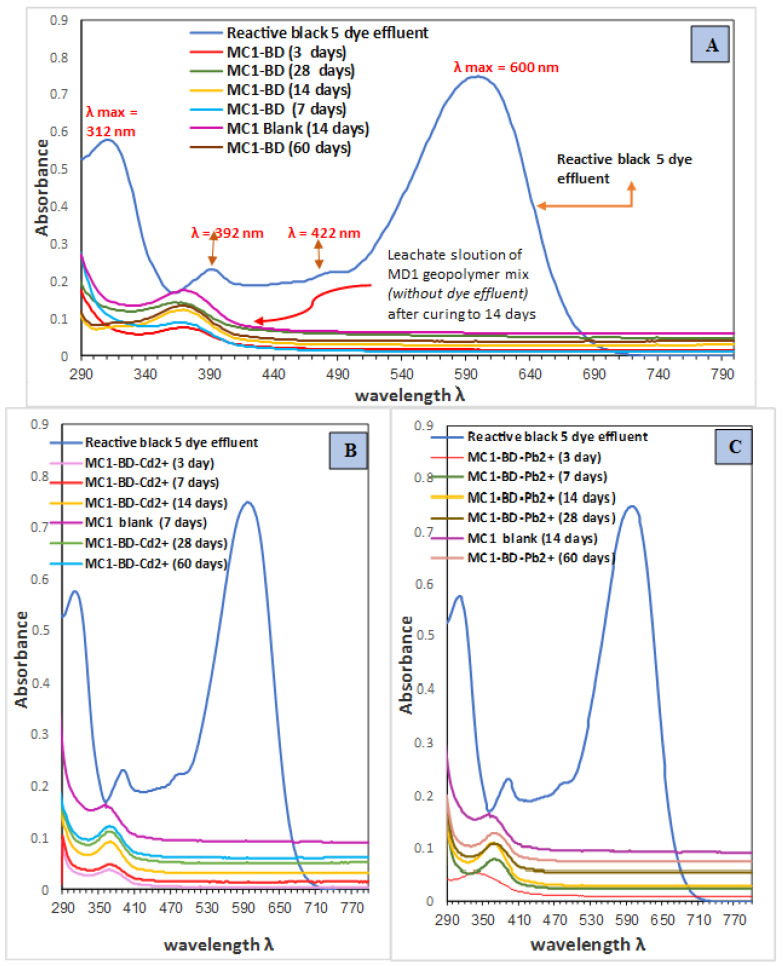
Absorption spectra of the reactive black 5 dye effluent and numerous leachate solutions for the following geopolymer mixes: (**A**) MC1-BD; (**B**) MC1-BD-Cd^2+^; and (**C**) MC1-BD-Pb^2+^ after various curing times up to 60 days.

**Figure 9 polymers-15-02865-f009:**
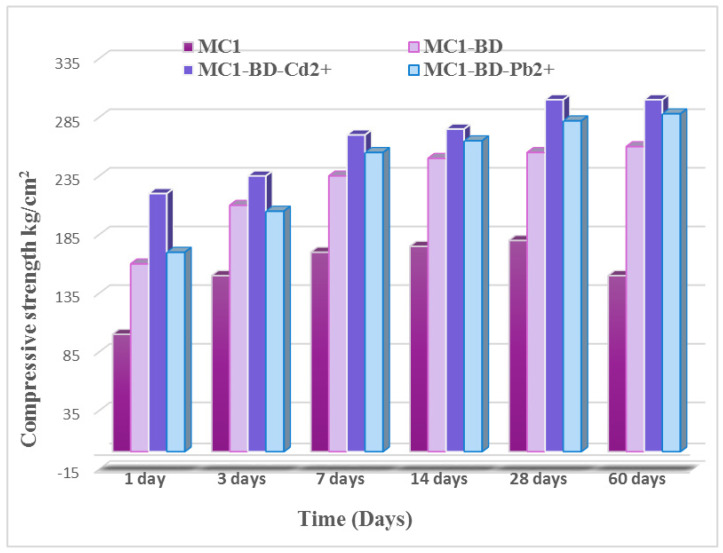
The compressive strength values of MC1, MC1-BD, MC1-BD-Cd^2+^, and MC1-BD-Pb^2+^ after curing at various periods until 60 days.

**Figure 10 polymers-15-02865-f010:**
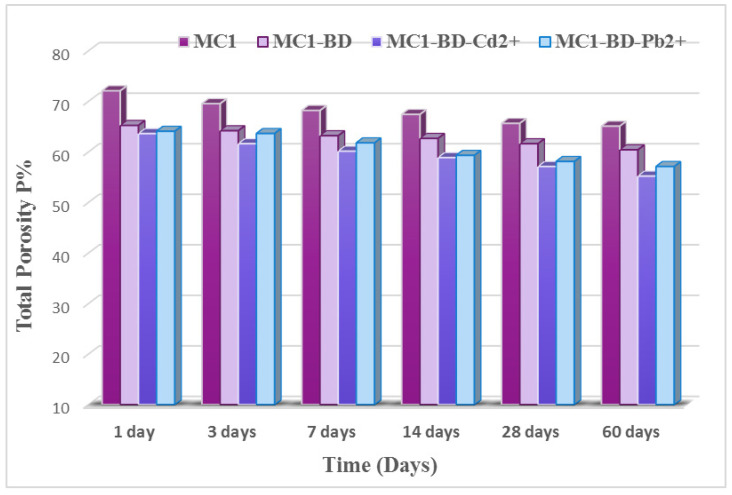
Total porosity % of MC1, MC1-BD, MC1-BD-Cd^2+^, and MC1-BD-Pb^2+^ after curing at various periods until 60 days.

**Figure 11 polymers-15-02865-f011:**
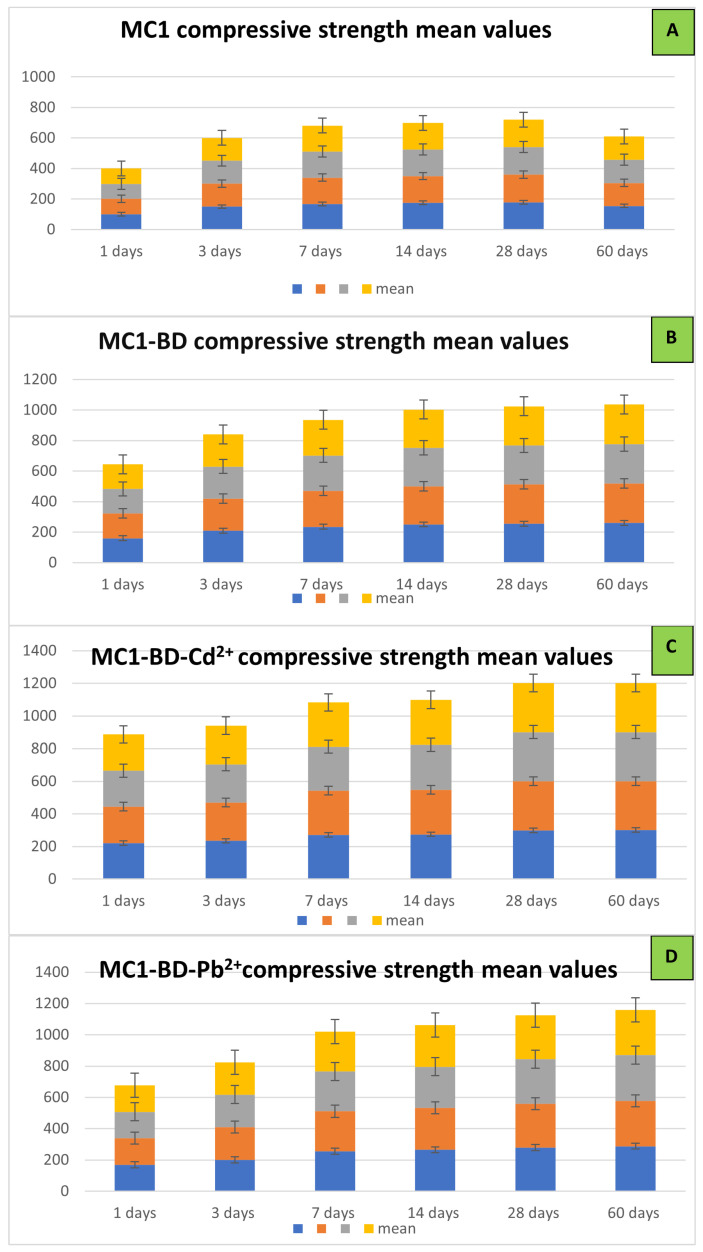
Compressive strength mean values for different MK/CKD mixes at different hydration times: (**A**) MC1, (**B**) MC1-BD, (**C**) MC1-BD-Cd^2+^, and (**D**) MC1-BD-Pb^2+^.

**Figure 12 polymers-15-02865-f012:**
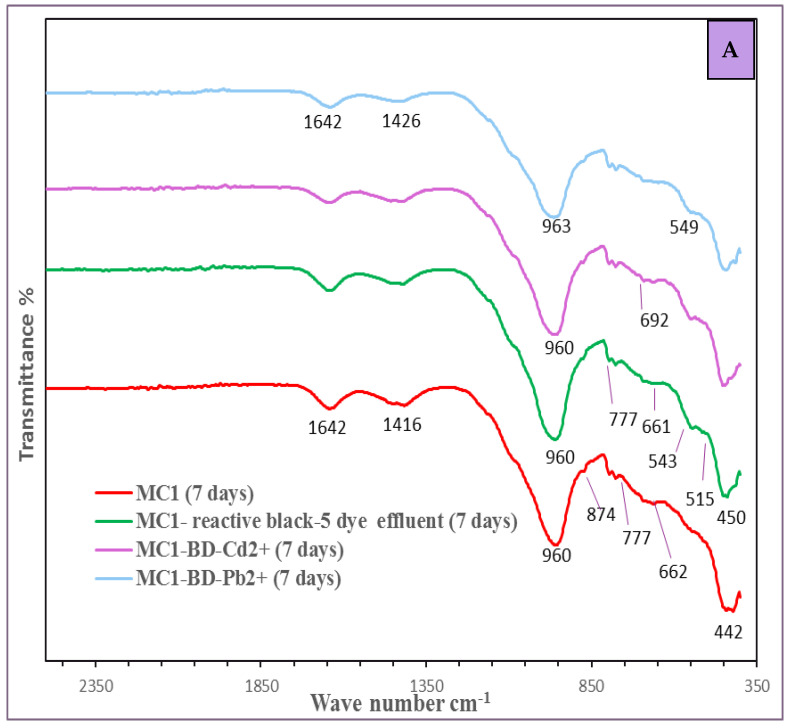
FTIR spectra of geopolymer pastes MC1, MC1-BD, MC1-BD-Cd^2+^ and MC1-BD-Pb^2+^ cured in water for (**A**) 7 days, (**B**) 28 days.

**Figure 13 polymers-15-02865-f013:**
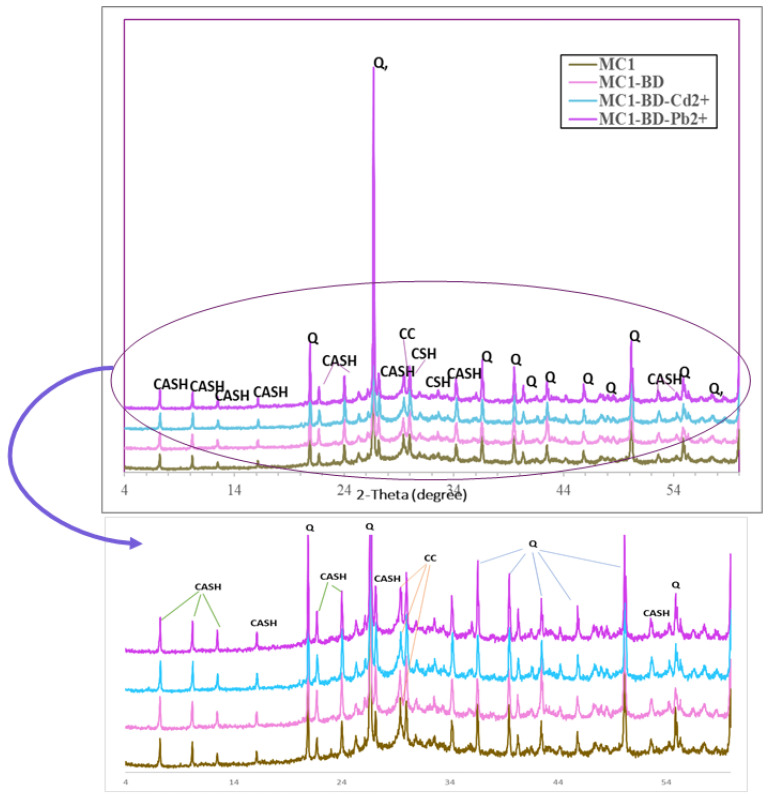
X-ray diffraction patterns of different MK/CKD-based geopolymer mixes under water curing for 28 days. **CASH = calcium aluminate silicate hydrate, CC = calcium carbonate, Q = quartaz.**

**Figure 14 polymers-15-02865-f014:**
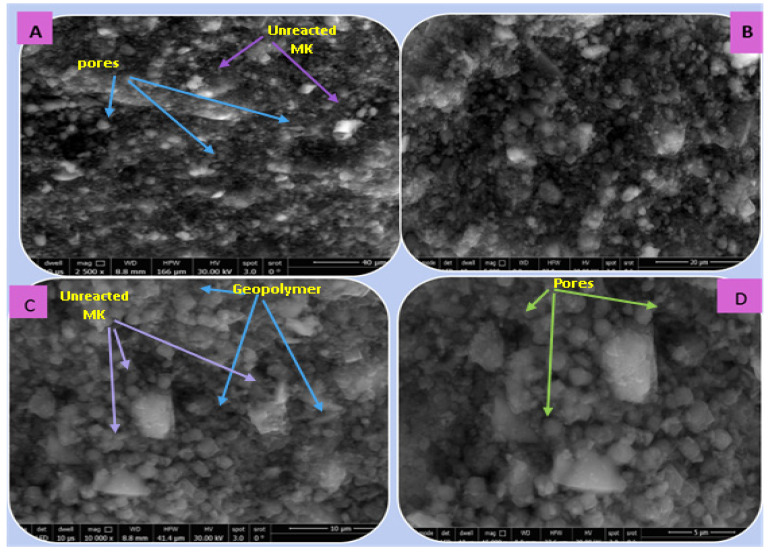
SEM micrographs of the geopolymer based on metakaolin incorporated with cement kiln dust (MC1) after 28 days of hydration at different magnifications: (**A**) 2500, (**B**) 5000, (**C**) 10,000, and (**D**) 15,000.

**Figure 15 polymers-15-02865-f015:**
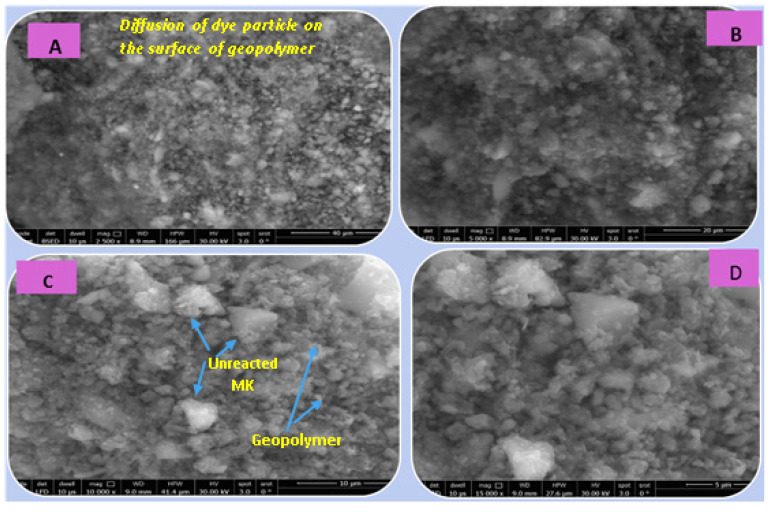
SEM micrographs of the geopolymer (MC1-BD) after 28 days of hydration at different magnifications: (**A**) 2500, (**B**) 5000, (**C**) 10,000, and (**D**) 15,000.

**Figure 16 polymers-15-02865-f016:**
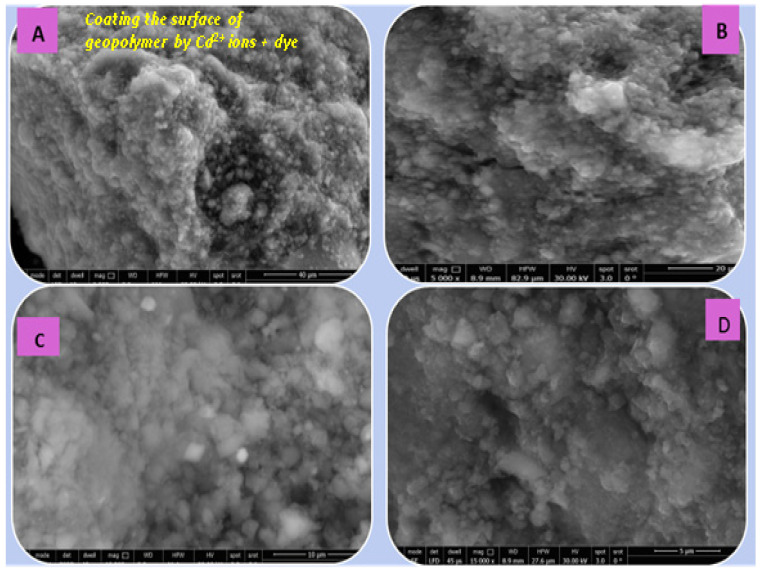
SEM micrographs of the geopolymer (MC1-BD-Cd2+) after 28 days of hydration at different magnifications: (**A**) 2500, (**B**) 5000, (**C**) 10,000, and (**D**) 15,000.

**Figure 17 polymers-15-02865-f017:**
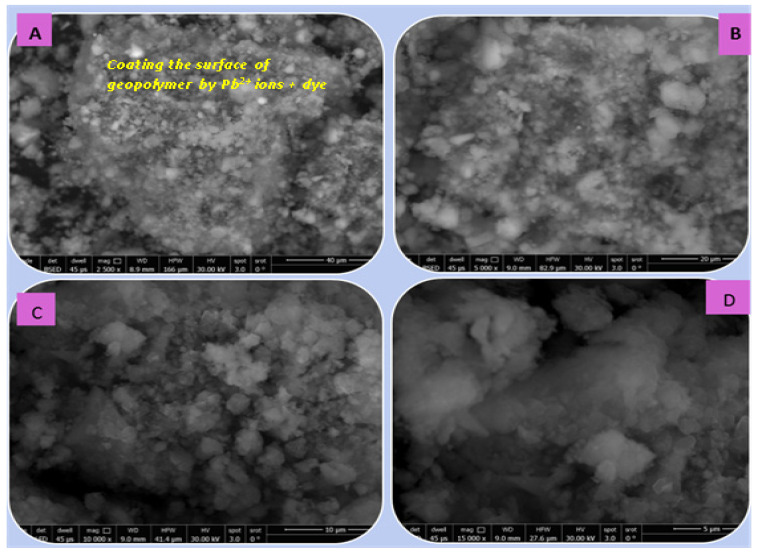
SEM micrographs of the geopolymer (MC1-BD-Pb2+) after 28 days of hydration at different magnifications: (**A**) 2500, (**B**) 5000, (**C**) 10,000, and (**D**) 15,000.

**Table 1 polymers-15-02865-t001:** Raw materials chemical oxide composition by XRF, mass %.

Type	Chemical Composition %											
	* SiO_2_ *	* Fe_2_O_3_ *	* CaO *	* Al_2_O_3_ *	* Cl^−^ *	* MgO *	* SO_3_ *	* Na_2_O *	* K_2_O *	* TiO_2_ *	* L.O.I *	* H_2_O *
**MK**	64.81	0.55	0.52	30.10	---	---	0.13	0.10	----	2.70	0.73	----
**CKD**	14.16	3.42	53.87	3.98	7.43	0.86	3.68	3.01	6.62	-----	2.80	-----
**LSS**	32.8	------	-----	-----	-----	-----	----	11.7	-----		-----	55.5

MK = Metakaolin, CKD = Cement kiln dust, **LSS =** Liquid sodium silicate.

**Table 2 polymers-15-02865-t002:** The chemical composition of the tested geopolymer mixes and their liquid/solid (L/S) ratios.

Mix Name	MK	CKD	Cd^2+^ Solution mL (1200 mg/L)	Reactive Black Dye Effluent mL	Pb^2+^ Solution mL (1200 mg/L)	NaOH wt.%	Na_2_SiO_3_ wt.%	L/S Ratio
**MC1**	80	20	---	---	---	15	15	0.52
**MC2**	60	40	---	---	---	15	15	0.60
**MC1-BD**	80	20	---	100	---	15	15	0.53
**MC1-BD-Cd^2+^**	80	20	 100	---	15	15	0.53
**MC1-BD-Pb^2+^**	80	20	---	 100	15	15	0.53

**Table 3 polymers-15-02865-t003:** The removal efficiency % values of the reactive black 5 dye effluent and heavy metal by MC1 and MC2 geopolymer mixes under optimum conditions using the adsorption approach.

Geopolymer Mix	Total Heavy Metals Concentration mg/L	Dye Effluent Removal %				Heavy Metal Ions Removal%	
		Before Adding Pb^2+^	After Adding Pb^2+^	Before Adding Cd^2+^	After Adding Cd^2+^	Pb^2+^	Cd^2+^
**MC1**	1200	46.3	22.9	46.3	54.7	47	7
**MC2**	1200	30.4	37.64	30.4	39.5	53	8.3

(wt. of adsorbent 0.01 g, temperature 30 °C, concentration of dye 35 mg/L, pH 2, time 120 min).

**Table 4 polymers-15-02865-t004:** Isotherm measurements for MC1 and MC2 geopolymer adsorbents.

Geopolymer Type	Langmuir Isotherm				Freundlich Isotherm		
	K_L_ (L/g)	a_L_ (L/mg)	Q_max_ (mg/g)	R^2^ (Correlation Coefficient)	K_F_	n	R^2^ (Correlation Coefficient)
**MC1**	2.673797	0.347594	7.692308	0.9911	2.8314	3.067485	0.9558
**MC2**	1.639344	0.362295	4.524887	0.9949	1.9634	3.968254	0.9254

**Table 5 polymers-15-02865-t005:** Leaching percent and immobilization values of reactive dye black 5 effluent in leachate solutions of different geopolymer mixes after different curing times (1, 3, 7, 14, 28, and 60 days).

Geopolymer Mixes	Leaching % & Immobilization	Time (Days)					
		1 Day	3 Days	7 Days	14 Days	28 Days	60 Days
MC1-BD	**Leaching %** Reactive black 5	0	0	0	0	0	0
	**Immobilization** Reactive black 5	100	100	100	100	100	100
**MC1-BD-Cd^2+^**	**Leaching %** **Cd^2+^**	0.0002011	0.0001720	0.0002586	0.0009195	0.0008621	0.0008621
	**Immobilization** **Cd^2+^**	**99.99979**	**99.99983**	**99.99974**	**99.99908**	**99.99914**	**99.99914**
	**Leaching %** Reactive black 5	0	0	0	0	0	0
	**Immobilization** Reactive black 5	100	100	100	100	100	100
**MC1-BD-Pb^2+^**	**Leaching %** **Pb^2+^**	0.002586	0.004023	0.0037356	0.0086206	0.005241	0.010344
	**Immobilization** **Pb^2+^**	**99.99741**	**99.99597**	**99.99626**	**99.99138**	**99.99475**	**99.98966**
	**Leaching %** Reactive black 5	0	0	0	0	0	0
	**Immobilization** Reactive black 5	100	100	100	100	100	100

**Table 6 polymers-15-02865-t006:** pH values of leachate solutions of MC1, MC1-BD, MC1-BD-Cd^2+^ and MC1-BD-Pb^2+^ curing in H_2_O at different hydration ages.

Geopolymer Mix	pH							
	0 h	1 h	1 Day	3 Days	7 Days	14 Days	28 Days	60 Days
**MC1**	7.05	12.64	13.22	13.29	13.44	13.45	13.49	13.58
**MC1-BD**	7.05	12.60	13.17	13.22	13.29	13.38	13.42	13.40
**MC1-BD-Cd^+2^**	7.04	12.54	13.29	13.15	13.28	13.43	13.40	13.42
**MC1-BD-Pb^2+^**	7.04	12.58	13.11	13.25	13.36	13.41	13.42	13.41

## Data Availability

Not applicable.

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
