# Peer review of "Green Synthesis of the Effectively Environmentally Safe Metakaolin-Based Geopolymer for the Removal of Hazardous Industrial Wastes Using Two Different Methods"

_polymers, 2023, doi:10.3390/polym15132865_

Round 1

Reviewer 1 Report

There are few comments for this study.

(1) The abstract is a short statement without notes or comments on the content of the paper. It is required to briefly explain the purpose of the research work, research methods and final main conclusions, etc

(2) The conclusion needs to be refined, please leave only the important findings.

(3) All the acronyms should be spelled out the first time they appear.

(4) The paper defines lots of acronyms, making the paper very hard to read. Please use leave the necessary acronyms. 

(5)     Please improve the illustrations/tables, with clear and of good quality.

The English can be improved.

Reviewer 2 Report

This work compares the efficacy of solidification and adsorption for reducing dye contaminants and heavy metals from wastewater using geopolymer based on metakaolin incorporation with cement kiln dust. Extensive experiments and characterizations have been conducted. The contribution is original and will attract broad interests. I recommend this paper can be accepted after minor revision.

1.     Will other ions affect the efficacy of geopolymer to adsorb Pb2+ or Cd2+?

2.     Is the adsorption of geopolymer reversible?

3.     The quality of figures should be improved.

Authers are suggested to polish the language throughout the manuscript to avoid inappropriate use.

Reviewer 3 Report

The manuscript entitled ‘Green Synthesis of the Effectively Environmentally Safe Metakaolin-based Geopolymer for the Removal of Hazardous Industrial Wastes Using Two Different Methods’ is in line with the Polymers journal. The topic of the article is up-to-date and important. This article is based on original research. Overall, the manuscript is properly composed; nevertheless, it requires some changes before publication, such as:

·       Abstract: try to make this part shorter.

·       Introduction (line 95): consider adding this examplea to introduction part: https://doi.org/10.3390/molecules28083520 and https://doi.org/10.3390/ma14216307

·       Introduction (last paragraph): stress the novelty of provided research

·       Chapter 2. (pkt. 2.2.3): give more details about geopolymer production, including temperatures

·       Chapter 2. (C3): add information about samples preparation

·       Chapter 3.2.2: Pls provide the results on MPa. Statistical analysis (min error bars) is required. How many samples have been tested in each series

·       All text should be formatted according to the template.

·       Lack of information on author contribution.

Minor editing of English language required

Round 2

Reviewer 1 Report

The authors have revised the manuscript as I suggested. I accepted their revisions.

Author Response

Thank you very much for your efforts and your time.

Reviewer 3 Report

The manuscript entitled ‘Green Synthesis of the Effectively Environmentally Safe Metakaolin-based Geopolymer for the Removal of Hazardous Industrial Wastes Using Two Different Methods’ was significantly improved, however there is still lack of statistical analysis for strenght results. It should be supplemented.

Minor editing of English language required
